# PDGFR-β signaling mediates HMGB1 release in mechanically stressed vascular smooth muscle cells

Ji On Kim[1,2‡], Seung Eun Baek[2‡], Eun Yeong Jeon[1,2], Jong Min Choi[2], Eun Jeong Jang[2], Chi Dae Kim[1,2]*

**1** Department of Pharmacology, School of Medicine, Pusan National University, Yangsan, Gyeongnam, Republic of Korea, **2** Gene & Cell Therapy Research Center for Vessel-associated Diseases, Pusan National University, Yangsan, Gyeongnam, Republic of Korea

‡ These authors have contributed equally to this work and share first authorship.
* chidkim@pusan.ac.kr

**Data Availability Statement:** All relevant data are within the manuscript and its Supporting Information files.

**Funding:** This research was supported by the Basic Science Research Program (NRF-

## Abstract

Mechanically stressed vascular smooth muscle cells (VSMCs) have potential roles in the development of vascular complications. However, the underlying mechanisms are unclear. Using VSMCs cultured from rat thoracic aorta explants, we investigated the effects of mechanical stretch (MS) on the cellular secretion of high mobility group box 1 (HMGB1), a major damage-associated molecular pattern that mediates vascular complications in stressed vasculature. Enzyme-linked immunosorbent assay (ELISA) demonstrated an increase in the secretion of HMGB1 in VSMCs stimulated with MS (0–3% strain, 60 cycles/min), and this secretion was markedly and time-dependently increased at 3% MS. The increased secretion of HMGB1 at 3% MS was accompanied by an increased cytosolic translocation of nuclear HMGB1; the acetylated and phosphorylated forms of this protein were significantly increased. Among various inhibitors of membrane receptors mediating mechanical signals, AG1295 (a platelet-derived growth factor receptor (PDGFR) inhibitor) attenuated MS-induced HMGB1 secretion. Inhibitors of other receptors, including epidermal growth factor, insulin-like growth factor, and fibroblast growth factor receptors, did not inhibit this secretion. Additionally, MS-induced HMGB1 secretion was markedly attenuated in PDGFR-β-deficient cells but not in cells transfected with PDGFR-α siRNA. Likewise, PDGF-DD, but not PDGF-AA, directly increased HMGB1 secretion in VSMCs, indicating a pivotal role of PDGFR-β signaling in the secretion of this protein in VSMCs. Thus, targeting PDGFR-β-mediated secretion of HMGB1 in VSMCs might be a promising therapeutic strategy for vascular complications associated with hypertension.

## Introduction

Increased blood pressure alters the blood vessels structurally and functionally, leading to reduced tissue perfusion and further cardiovascular complications [1]. The prolonged vasoconstriction in a resistant artery has been considered as the main driver of vascular remodeling in hypertension [2]. In response to increased blood pressure, small resistance vessels undergo vascular remodeling

2020R1A2C1005135) and the Medical Research Center (MRC) Program (NRF-2015R1A5A2009656) through the National Research Foundation of Korea (NRF) grant funded by the Korean government (MSIP). The funders had no role in study design, data collection and analysis, decision to publish, or preparation of the manuscript.

**Competing interests:** The authors have declared that no competing interests exist.

**Abbreviations:** VSMC, Vascular smooth muscle cell; HMGB1, High mobility group box 1; MS, Mechanical stretch; DAMP, Damage-associated molecular pattern; PDGFR, Platelet-derived growth factor receptor; ELISA, Enzyme-linked immunosorbent assay; EGF, Epidermal growth factor; IGF, Insulin-like growth factor; FGF, Fibroblast growth factor.

[3]. The walls of the vasculatures during remodeling become thicker, stiffer and less elastic, thereby increasing the risk of vascular obstruction and subsequent organ failure [4,5].

Alterations in the regulation of vascular tone have been described previously among various risk factors associated with cardiovascular complications [6]. The physical force exerted by blood pressure continuously exposes the vascular wall to mechanical stretch. The higher the blood pressure, higher the force of stretch, leading to vascular hypertrophy and remodeling [7–9]. Vascular smooth muscle cells (VSMCs) are essential regulators of mechanical forces, and excessive hemodynamic forces in arterial hypertension leads to mechanical stretch (MS) of VSMCs [10].

Reportedly, MS increases VSMC growth via activation of the epidermal growth factor (EGF) receptor [11] and induces its proliferation via the insulin-like growth factor (IGF) receptor [12] and platelet-derived growth factor (PDGF) receptor (PDGFR) [13]. In previous studies, PDGF was reported to have the most important implications for susceptibility to hypertension among various growth factors [14]. As described previously by Raines [15], PDGF is a family of growth factors consisting of disulfide-bonded homo- and heterodimers encoded by four genes, PDGF-A, PDGF-B, PDGF-C, and PDGF-D. PDGF acts on cells by binding to either homodimer or heterodimer of the two PDGFRs, namely PDGFR-α and PDGFR-β. However, the individual role of PDGFR isoforms in the pathogenesis of vascular complications in hypertension has not been fully elucidated.

The molecules released from mechanically stressed vascular cells have been suggested as mediators that induce vascular complications in hypertension [16]. Among various damage-associated molecular patterns (DAMPs), high mobility group box 1 (HMGB1) has been suggested as an important regulator of inflammation in the injured vasculatures [17–19]. HMGB1 is known to engage in a variety of cellular activities in the nucleus [20]. In addition to the activities in the nucleus, this molecule behaves as a key DAMPs that mediate a number of cellular reactions via translocation of HMGB1 into the cytoplasm and its release into the extracellular medium [21–25]. In atherosclerotic lesions, the high concentrations of HMGB1 have been demonstrated, suggesting a potential role for HMGB1 in vascular remodeling through increased proliferation and migration of vascular cells [26]. Although the major cells of the vascular walls are VSMCs, the precise role of these cells in inducing HMGB1 secretion in the stressed vasculature is unclear.

Given the importance of PDGFR signaling in VSMCs, we hypothesized that the MS-PDGFR signaling axis would directly influence the secretion of HMGB1 in MS-stimulated VSMCs. Therefore, in this study, we investigated the effects of MS on the cellular secretion of HMGB1 and clarified the molecular mechanisms involved in the secretion of HMGB1 in MS-stimulated VSMCs.

## Materials and methods

### Ethics statements and animals

All animal procedures conformed with the Guide for the Care and Use of Laboratory Animals published by the US National Institutes of Health (NIH Publication No.85-23, 2011 revision), and all experimental protocols were reviewed and approved by the Pusan National University Institutional Animal Care and Use Committee. Sprague-Dawley rats were purchased from Charles River Breeding Laboratories (Kingston, NY, USA).

### Chemicals and antibodies

HMGB1 antibody (10829-1-AP) was purchased from Proteintech (Chicago, IL, USA). PDGFR-α (3164S), PDGFR-β (3169S) and phospho-specific (2992S, 3166S) antibodies were

obtained from Cell Signaling Technology (Beverly, MA, USA). β-Actin (sc-47778) and Lamin B1 (sc-374015) antibodies were purchased from Santa Cruz Biotechnology Inc. (Dallas, TX, USA). NE-PERTM reagents (78835) were purchased from Thermo Fisher Scientific (Rockford, IL, USA). Puredown protein G-agarose (P9202-050) was purchased from GenDEPOT (Barker, TX, USA). IP lysis buffer (87787) was purchased from Thermo Fisher Scientific. Various growth factor receptor inhibitors such as AG1024 (65678-07-1; an IGF receptor inhibitor), AG1295 (71897-07-9; a PDGF receptor inhibitor), AG1478 (175178-82-2; an EGF receptor inhibitor), and PD173074 (219580-11-7; an FGF receptor inhibitor) were purchased from Calbiochem (La Jolla, CA, USA). Recombinant PDGF-AA (221-AA) and-DD (1159-SB) were purchased from R&D Systems Inc. (Minneapolis, MN, USA). Horseradish peroxidase (HRP)-conjugated IgG secondary antibodies were purchased from Santa Cruz Biotechnology Inc.

## Cell culture

Sprague-Dawley rats (7 weeks old, male) were euthanized by $CO_2$ inhalation and dissected to separate the thoracic aorta. Endothelial cells of the excised aorta were physically peeled off by scraping and the aorta was cut into approximately 1 mm$^2$ segments. Each segment of the aorta was covered with glass and explanted in a cell culture dish for a week. Primary VSMCs were cultured in Dulbecco's modified Eagle medium (DMEM; Gibco BRL, Grand Island, NY, USA) with 10% fetal bovine serum (FBS; Gibco BRL) and antibiotic-antimycotic solution (Gibco BRL). Cells were then maintained at 37˚C in a humidified 5% $CO_2$/95% air atmosphere.

## Mechanical stretch

To stimulate VSMCs with MS, cells were seeded onto flexible-bottomed 6-well BioFlex culture plates (Dunn Labortechnik, Germany; BF-3001C). The cells were incubated in $CO_2$ incubator at 37˚C, 95% humidity and 5% $CO_2$ for 24 hrs. When cells reached confluency, the medium was replaced with serum-free medium and the cells were stimulated with MS. A Flexercell Tension Plus FX-4000T system (Flexcell International Corp., Hillsborough, NC, USA) was used to apply physiological equibiaxial cyclic stretch (1 Hz, 0–3% strain, 60 cycles/min, 0–12 hrs).

## Western blot analysis

VSMCs were lysed in ice-cold lysis buffer (Thermo Fisher Scientific). Equal amounts of protein were separated on 8–10% polyacrylamide gels under reducing conditions, and then transferred onto nitrocellulose membranes (Amersham Pharmacia Biotech, Piscataway, NJ, USA). Membranes were blocked with 5% skim milk in Tris-buffered saline with Tween-20 (TBST) at room temperature for 2 hrs and then incubated overnight with the primary antibody at 4˚C. The incubated membranes were then washed with TBST and incubated with HRP-conjugated secondary antibody at room temperature for 2 hrs. After washing again with TBST, blots were developed using enhanced chemiluminescence (ECL) western blotting detection reagents (Thermo Fisher Scientific). Membranes were re-probed with anti- β -actin antibody to detect β-actin that was used as an internal control. Protein signals were quantified using the UN-SCAN-IT GEL 7.12 program, and data were expressed as relative β-actin densities.

## Immunoprecipitation (IP)

The beads were collected from Puredown protein G-agarose by centrifugation for 1 min at 7000 rpm and washed with IP lysis buffer two to three times. The final wash was gently removed, and HMGB1 antibody was added to the beads. The bead-antibody mixture was

rotated at 4˚C for 2 days and then washed with IP lysis buffer. The captured antigen was collected by centrifugation for 1 min at 7000 rpm, and then resuspended in IP lysis buffer containing protease inhibitors and added to the protein sample. After rotated at 4˚C for 2 hrs, the samples were centrifuged at 7000 rpm for 1 min and the supernatant was gently removed. IP lysis buffer with protease inhibitors was added again. And then it boiled in a heat block for 5 min, incubated on ice for 5 min, and centrifuged at 12000 rpm for 2 min. The final supernatant was gently gathered and used for Western blot analysis.

### Small interfering RNA (siRNA) preparation and transfection

PDGFR-α and PDGFR-β siRNA oligonucleotides were synthesized by Bioneer. The siRNA negative control duplex was used as a control. All siRNA molecules were transfected using Lipofectamine 2000 (Invitrogen, Carlsbad, CA, USA). For siRNA transfection, VSMCs were seeded in 6-well plates and grown for 24 hrs. Next, after replacing the culture medium with Optimized-Minimal Essential Medium (Opti-MEM; Gibco BRL), the cells were transfected with siRNA for PDGFR-α or PDGFR-β and negative control using Lipofectamine 2000 and incubated for 6 hrs at 37˚C. Then, Opti-MEM was replaced with DMEM, and the cells were incubated at 37˚C for 48 hrs.

### Enzyme-linked immunosorbent assay (ELISA)

HMGB1 secretion was measured in the culture medium of rat VSMCs stimulated with MS using the rat HMGB1 ELISA kit (Elabscience, Houston, TX, USA; E-EL-R0505) according to the manufacturer's instructions.

### Statistical analysis

Results are expressed as the mean ± standard error of mean (SEM). The result 4 and 6 used one-way analysis of variance (ANOVA) followed by Dunnett multiple comparison test and result 1, 2, 3, 5, 7 used student's t-test to determine significant differences. Statistical significance was accepted at P values $< 0.05$.

## Results

### Increased HMGB1 secretion in MS-stimulated VSMCs

To determine the effects of MS on HMGB1 secretion in VSMCs, cells derived from rat thoracic aorta explants were cultured and seeded onto BioFlex 6-well culture plates and then stimulated with 0–3% MS for 0–12 hrs. The secretion of HMGB1 in MS-stimulated cells was markedly elevated compared with that in non-stimulated cells. When 3% strain was applied to VSMCs, the secretion of HMGB1 constantly increased until 12 hrs ($52.57 \pm 5.24$-fold, $^{**}$p $< 0.01$), but the time dependency was not observed in 1% MS. Since 3% strain showed the most substantial increase in HMGB1 secretion in this study, this level of strain was used for the subsequent experiments (Fig 1).

### Increased cytosolic translocation of nuclear HMGB1 in the MS-stimulated VSMCs

The cytosolic translocation of nuclear HMGB1 is the first prerequisite for the secretion of HMGB1 in VSMCs. To investigate the effect of MS on cytosolic translocation of nuclear HMGB1 in VSMCs, cells were stimulated with 3% MS for 0–1 hr, and then separated into nucleus and cytosol fractions. In the Western blot analysis for HMGB1 concentration, the nuclear fraction of HMGB1 was markedly decreased ($0.42 \pm 0.10$-fold, $^{**}$p $< 0.01$), while

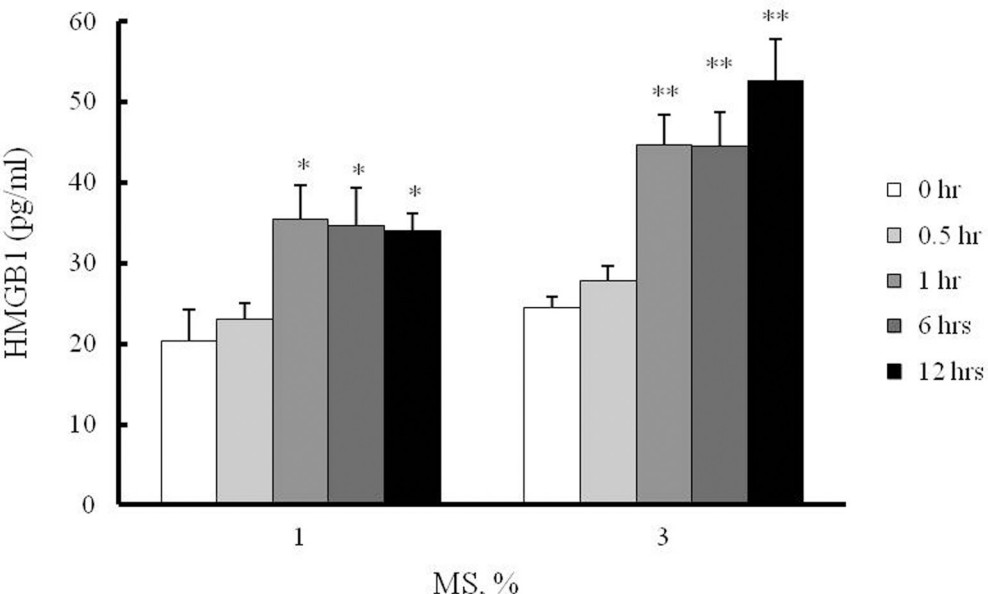

**Fig 1. Time-course and force-dependent effects of MS on HMGB1 secretion in VSMCs.** Rat aortic VSMCs were stressed with Flexercell Tension Plus FX-4000T system (1 and 3% strain, 60 cycles/min) for 0–12 hrs. HMGB1 secreted into the cell culture media was measured by ELISA, and results were expressed as the mean ± SEM of 6 independent experiments. $*P < 0.05$ and $**P < 0.01$ vs. corresponding value in 0 hr.

cytosolic fraction was gradually increased up to 1 hr of 3% MS (2.97 ± 0.34-fold, $**$p < 0.01) (Fig 2).

## Increases in acetylated- and phosphorylated HMGB1 in VSMCs stimulated with MS

The translocation of HMGB1 during cell activation is mediated by acetylation and phosphorylation processes, and is prerequisite for the secretion of HMGB1 [27]. To evaluate the

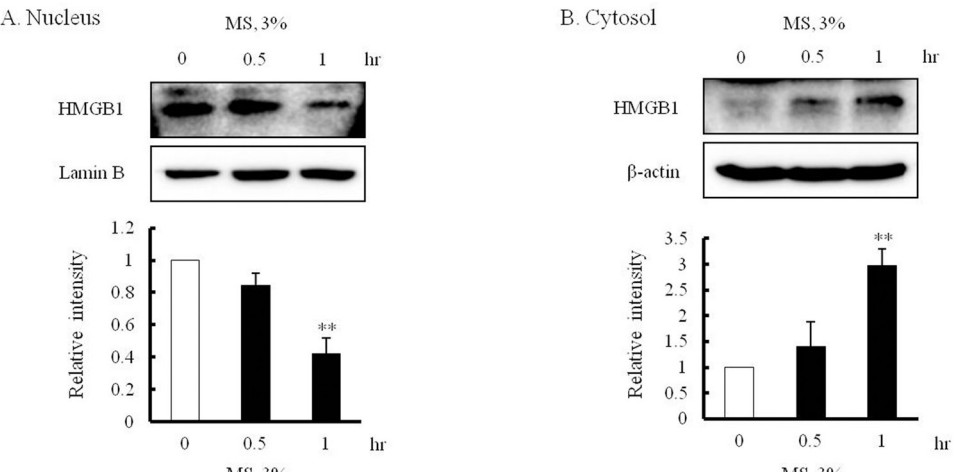

**Fig 2. Effects of MS on translocation of nuclear HMGB1 into cytosol in VSMCs.** VSMCs were stimulated by 3% MS for 0–1 hr, and then HMGB1 in the cytosolic and nuclear fractions were measured by Western blot. Lamin B1 and β-actin were used as internal controls for nuclear and cytosolic HMGB1, respectively. Quantitative results were expressed as the means ± SEMs of 3–4 independent experiments. $**P < 0.01$ vs. corresponding value in 0 hr.

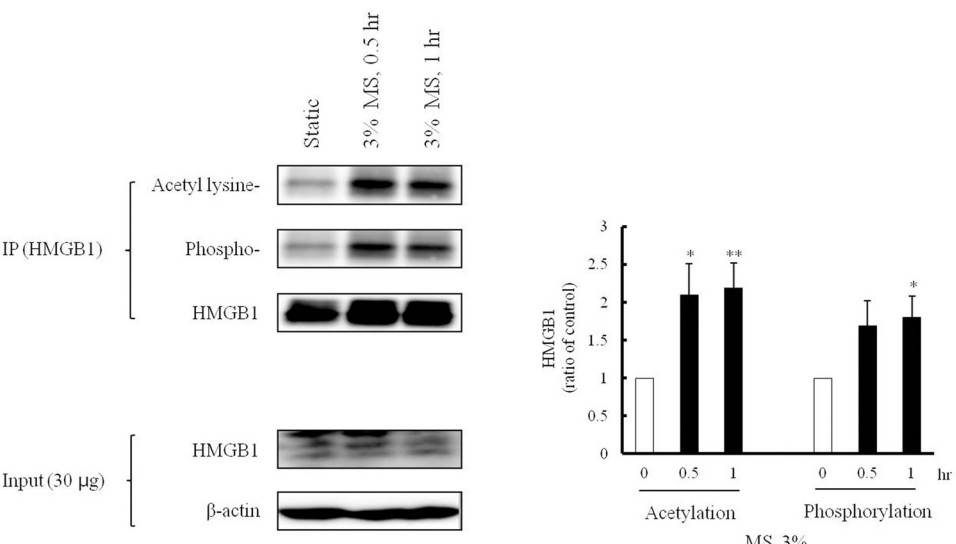

**Fig 3. Effects of MS on acetylation and phosphorylation of HMGB1 in VSMCs.** VSMCs were stimulated with 3% MS for 0–1 hr, and HMGB1 was immunoprecipitated (IP), and then acetylated and phosphorylated HMGB1 was measured by Western blot. HMGB1 and β-actin were used as internal controls for IP and Input HMGB1, respectively. Quantitative results were expressed as the means ± SEMs of 4 independent experiments. $^*P < 0.05$ vs. corresponding value in 0 hr.

involvement of these processes of HMGB1 in the cytosolic translocation, cells were stimulated by 3% MS for 0–1 hr, and the protein of the lysed cell was combined with HMGB1 antibody and then immunoprecipitated. In Western blot analysis for HMGB1, acetylated- and phosphorylated HMGB1 were significantly increased in VSMCs stimulated with 3% MS. Acetylated- and phosphorylated HMGB1 showed a different tendency, that acetylated- HMGB1 increased until 1 hr (2.27 ± 0.42-fold, $^{**}$p < 0.01), while phosphorylated HMGB1 did not increased until 0.5 hr. These results suggest a potential involvement of acetylation and phosphorylation processes in the cytosolic translocation of nuclear HMGB1 in MS-stimulated VSMCs (Fig 3).

## HMGB1 secretion via PDGFR signaling in MS-stimulated VSMCs

To identify the mechanoreceptor mediating HMGB1 secretion in the MS-stimulated VSMCs, cells were pretreated for 1 hr with various inhibitors for the potential mechanoreceptors, and then stimulated with 3% MS for 1 hr. As shown in Fig 4, HMGB1 secretion induced by 3% MS was markedly attenuated by AG1295, a PDGF receptor inhibitor (28.01 ± 1.93-fold, $^{##}$p < 0.01), but not by others including AG1024 (an IGF receptor inhibitor) (46.87 ± 0.88-fold), AG1478 (an EGF receptor inhibitor) (46.85 ± 2.52-fold), and PD173074 (a FGF receptor inhibitor) (40.00 ± 1.82-fold). Thus, it was suggested that the signaling mediated by PDGF receptor might play a pivotal role in MS-induced HMGB1 secretion in VSMCs (Fig 4).

## Increased phosphorylation of PDGFR in MS-stimulated VSMCs

To identify the individual role of PDGFR isoforms in the MS-induced HMGB1 secretion in VSMCs, cells were stimulated with 3% MS for 1 hr, and then PDGFR phosphorylation was determined by Western blot analysis. As shown in Fig 5A, the phosphorylation of PDGFR-α and PDGFR-β in 3% MS-induced cells was markedly increased, which was accompanied by an increased expression of both PDGFR-α and PDGFR-β. To evaluate the involvement of

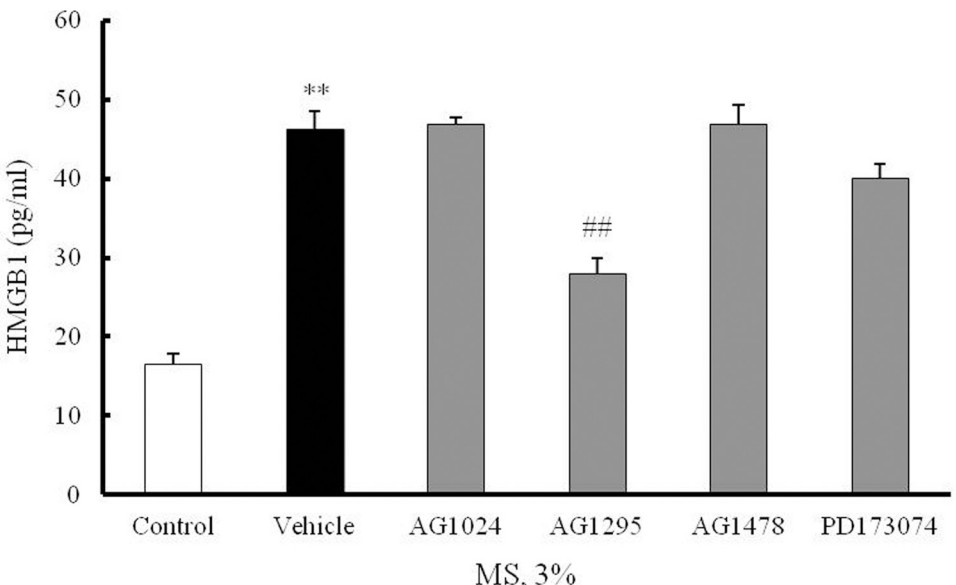

**Fig 4. Involvement of PDGFR signaling on HMGB1 secretion in MS-stimulated VSMCs.** VSMCs were pretreated with inhibitors for various growth factor receptors including AG1024 (10 μm, an IGF receptor inhibitor), AG1295 (10 μm, an PDGF receptor inhibitor), AG1478 (10 μm, an EGF receptor inhibitor), and PD173074 (10 μm, an FGF receptor inhibitor), and then stimulated with 3% MS for 1 hr. HMGB1 secreted into the cell culture media was measured by ELISA, and results were expressed as the mean ± SEM of 6 independent experiments. $^{**}P < 0.01$ vs. corresponding value in control, $^{##}P < 0.01$ vs. corresponding value in vehicle.

PDGFR expression on an increased phosphorylation of PDGFR, the ratios of phosphorylated PDGFR to both PDGFR and β-actin were determined as shown in Fig 5B. The ratios (3.65 ± 0.39-fold, $^{#}p < 0.05$ in PDGFR-β) of phosphorylated PDGFRs to β-actin were higher than those (2.20 ± 0.28-fold, $^{#}p < 0.05$ in PDGFR-β) of phosphorylated PDGFRs to PDGFR, indicating the possible involvement of both an increased receptor expression and activated phosphorylation signal on MS-induced phosphorylation of PDGFR (Fig 5).

## Role of PDGFR-β signaling in MS-induced HMGB1 secretion in VSMCs

To further investigate the role of PDGFR isoform, VSMCs deficient of PDGFR-α or PDGFR-β were produced by transfection of their specific siRNA, and then stimulated with 3% MS for 1 hr. In cells transfected with 200 nM of siRNA, the expression of PDGFR-α (1.20 ± 0.22-fold, $^{#}p < 0.05$) and PDGFR-β (1.40 ± 0.46-fold, $^{#}p < 0.05$) were markedly attenuated. In cells deficient of PDGFR-β, the increased secretion of HMGB1 in 3% MS stimulated VSMCs was markedly attenuated (132.53 ± 13.94-fold, $^{##}p < 0.01$), but not in PDGFR-α-deficient cells (196.95 ± 13.00-fold) (Fig 6).

## The direct evidence of PDGFR-β involvement in HMGB1 secretion in VSMCs

To define the individual role of PDGFRs in HMGB1 secretion in VSMCs, cells were stimulated with PDGF-AA (0–5 ng/ml) and PDGF-DD (0–5 ng/ml) for 10 min, and then HMGB1 secretion was determined by ELISA analysis. As expected, the phosphorylation of PDGFR-α and PDGFR-β was markedly increased by PDGF-AA (1 and 5 ng/ml) (1.93 ± 0.08-fold, $^{**}p < 0.01$) and PDGF-DD (1 and 5 ng/ml) (6.07 ± 0.25-fold, $^{**}p < 0.01$), respectively. In cells treated with PDGF-DD, HMGB1 secretion was markedly increased (42.51 ± 1.53-fold, $^{**}p < 0.01$),

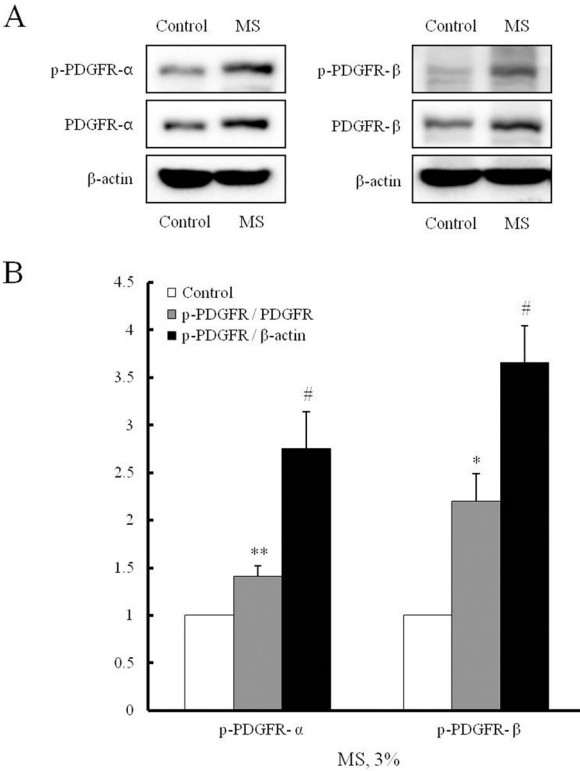

**Fig 5. Effects of MS on phosphorylation of PDGFR in VSMCs.** VSMCs were stimulated with 3% MS for 1 hr, and the expression levels of phosphorylated and total PDGFR isoforms (PDGFR-α and PDGFR-β) were measured by Western blot. β-Actin was used as internal controls for total and phosphorylated PDGFR isoforms. Quantitative results were expressed as the mean ± SEM of 3–4 independent experiments. **P < 0.01 vs. corresponding value in control, #P < 0.05 vs. corresponding value in p-PDGFR/PDGFR.

while HMGB1 secretion was not induced in cells treated with PDGF-AA (18.81 ± 1.15-fold). These results indicate that the PDGFR-β signaling pathway play a critical role on the regulation of HMGB1 secretion in VSMCs (Fig 7).

## Discussion

An increased vascular tone is a pivotal pathogenic event mediating vascular complications in hypertension. In this study, the mechanical stretch (MS) on VSMCs, which mimics increased pressure in the vasculatures, showed an increase in the release of HMGB1, a major DAMP implicated in vascular inflammation. The MS-mediated release of HMGB1 was attenuated in VSMCs deficient of PDGFR-β as well as in cells pretreated with PDGFR inhibitor. Moreover, the increased release of HMGB1 was demonstrated in cells treated with PDGF-DD, suggesting a pivotal role for PDGFR-β signaling in the increased secretion of HMGB1 in MS-stimulated VSMCs.

The phenotypes of VSMCs are influenced by diverse hormonal and environmental factors, including cytokine stimulation, cell-cell contact, cellular adhesion, vascular injury, and increased mechanical force [28]. In the blood vessels, the intravascular pressure is transmitted to the arterial walls and counterbalanced by mechanical stress, which stretches the circumferentially oriented VSMCs [2,6,10]. VSMCs in the vasculatures are constantly subjected to mechanical forces as a consequence of pulsatile blood flow and shear stress. Among multiple hemodynamic forces, VSMCs are primarily subjected to pulsatile cyclic stretch in response to

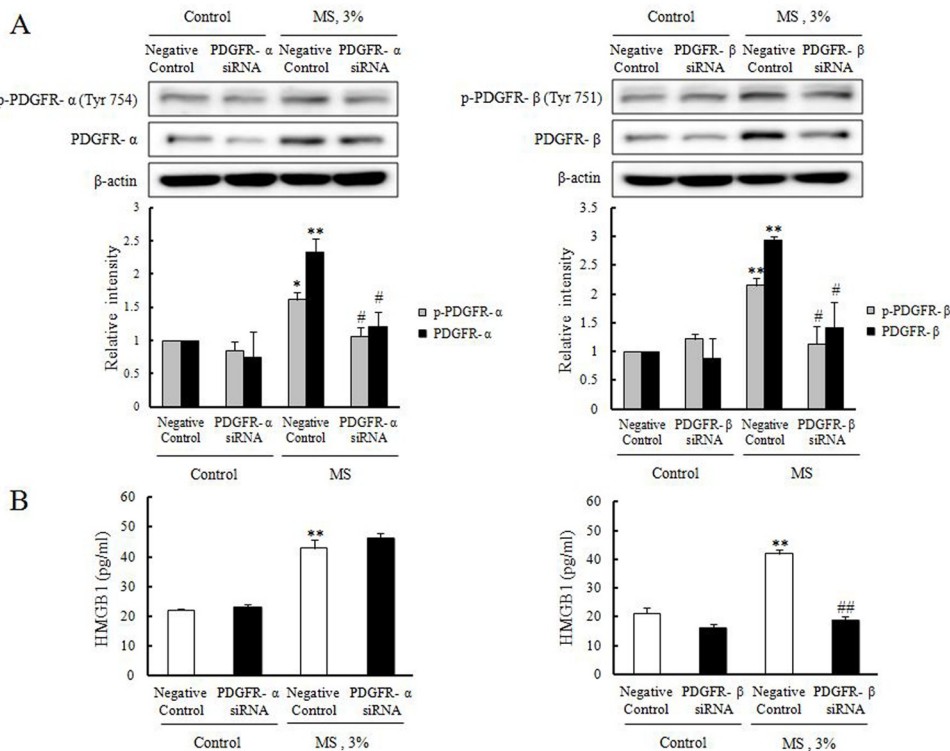

**Fig 6. HMGB1 secretion via PDGFR-β signaling pathway in MS-stimulated VSMCs. (A)** VSMCs were transfected with PDGFR-α or PDGFR-β siRNA (200 nM) for 48 hrs, and then stimulated by 3% MS for 1 hr. The expression levels of phosphorylated and total PDGFR isoforms (PDGFR-α and PDGFR-β) were determined by Western blotting using β-actin as an internal control. Quantitative results were expressed as the mean ± SEM of 3–4 independent experiments. $^{*}P < 0.05$ and $^{**}P < 0.01$ vs. corresponding value in control, $^{#}P < 0.05$ vs. corresponding value in negative control. **(B)** VSMCs were transfected with PDGFR-α or PDGFR-β siRNA (200 nM) for 48 hrs, and then stimulated by 3% MS for 1 hr. HMGB1 secreted into the culture media was determined by ELISA, and results were expressed as the mean ± SEM of 6 independent experiments. $^{**}P < 0.01$ vs. corresponding value in control, $^{##}P < 0.01$ vs. corresponding value in vehicle.

systolic-diastolic fluctuations in blood pressure. The higher the blood pressure, the higher the force of MS, leading to vascular hypertrophy and remodeling [29]. Although vascular remodeling is a compensatory mechanism to hypertension, it is detrimental because it structurally and functionally changes the blood vessels, leading to reduced tissue perfusion and further inducing hypertension [30]. However, the precise role of VSMCs in vascular alterations in hypertension is unclear.

Although the mechanism by which physical factors such as hypertension causes these pathological changes in the blood vessels has not yet been determined, several possible initial mediators have been considered. Reportedly, HMGB1 is known as a multifunctional protein which induces vascular remodeling in hypertension via phenotypic transformation of VSMCs from contractile to synthetic type [31–33]. The released extracellular HMGB1, a key damage-associated molecular pattern (DAMP) molecule, is the central mediator of lethal inflammation in tissue damage or infection [34]. In the development and progression of cardiovascular diseases, HMGB1 is one of the best characterized DAMPs among various injury-induced mediators [35]. Moreover, HMGB1 levels in atherosclerotic plaque were increased, suggesting a pivotal role for HMGB1 in the process of vascular remodeling via the potentiation of inflammatory processes [36].

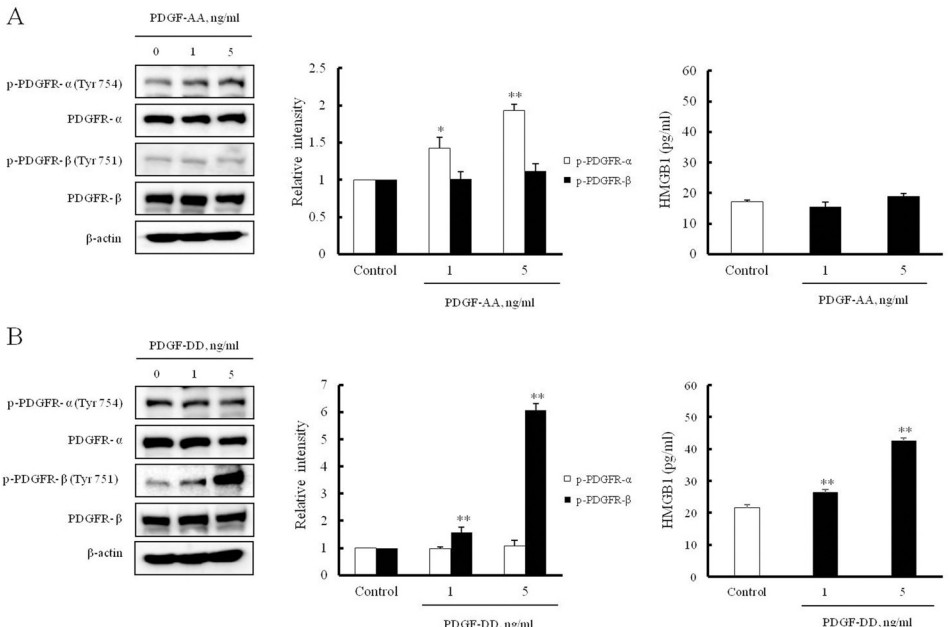

**Fig 7. The individual role of PDGF-AA and PDGF-DD on the secretion of HMGB1 in VSMCs.** VSMCs were stimulated with the indicated doses of PDGFR-α ligand (PDGF-AA) and PDGFR-β ligand (PDGF-DD) for 10 min, and then the levels of phosphorylated and total PDGFR-α and PDGFR-β isoforms were determined by Western blotting. β-Actin was used as an internal control. Quantitative results for the expression of PDGFR-α and PDGFR-β were expressed as the mean ± SEM of 3–4 independent experiments. *$P < 0.05$ and **$P < 0.01$ vs. corresponding value in control. HMGB1 secreted into the culture media was measured by ELISA, and results were expressed as the mean ± SEM of 6 independent experiments. *$P < 0.05$ and **$P < 0.01$ vs. corresponding value in control.

To evaluate the effect of high pressure on the secretion of HMGB1, VSMCs were stimulated with 0–3% strain, 1 Hz MS mimicking an increased pressure in the vasculatures. VSMCs cultured from rat thoracic aorta were seeded onto 6 well Bioflex cell culture plates. The cultured VSMCs were starved with 0.5% FBS for 24 hrs, and then stimulated with 1–3% MS for 0–12 hrs. Compared to non-stimulated cells, the MS-stimulated cells showed an elevation in HMGB1 secretion in association with the increased cytosolic translocation of nuclear HMGB1. Moreover, 3% MS showed an increased acetylation and phosphorylation of nuclear HMGB1, indicates that the observed increase in HMGB1 release in MS-stimulated VSMCs was directly induced by the upregulated acetylation and phosphorylation of HMGB1. However, whether the observed increase in HMGB1 release in MS-stimulated VSMCs was directly induced by the upregulated expression of HMGB1 has not been elucidated yet.

It has been shown that there are various mechanosensors in the vascular cell membrane, including proteins such as integrins [37], G proteins and G protein-coupled receptors [38], receptor tyrosine kinase [39], and calcium channel [37] and intercellular junction proteins [40]. In addition, growth factors such as platelet-derived growth factor (PDGF), fibroblast growth factor (FGF) and transforming growth factor-beta (TGF-β) have been suggested to be involved in some phases of the pathologic changes [41]. To identify the mechanoreceptor that induce HMGB1 secretion in MS-stimulated VSMCs, cells were pretreated with various growth factor receptor inhibitors, and then exposed to 3% MS. As shown in Fig 4, HMGB1 secretion was significantly inhibited by AG1295, a PDGF receptor inhibitor, but not by other inhibitors for IGF, EGF, and FGF receptors, suggesting a role of PDGF receptor signaling on HMGB1 secretion in VSMCs.

It is known that PDGFs are composed of A and B polypeptide chains, which form the three different disulfide-bound dimer proteins PDGF-AA, PDGF-BB, and PDGF-AB. Two different plasma membrane receptors for PDGFs have been identified. PDGF receptor α (PDGFR-α) binds all forms of PDGFs, whereas PDGF receptor β (PDGFR-β) binds PDGF-BB and, to a lesser extent, PDGF-AB, but not PDGF-AA [42]. In the present study, both the phosphorylation of PDGFR-α and PDGFR-β was markedly increased in cells stimulated with 3% MS, suggesting MS-induced autophosphorylation of tyrosine residues in the PDGFR kinase domain [43]. Interestingly, the MS-induced HMGB1 secretion was significantly attenuated in PDGFR-β-deficient cells, but not in cells transfected with siRNA for PDGFR-α. Likewise, PDGF-DD, but not PDGF-AA, directly increased HMGB1 secretion in VSMCs, indicating a pivotal role of PDGFR-β signaling on MS-induced HMGB1 secretion in VSMCs.

Considering our results in which MS increased HMGB1 release via acetylation and phosphorylation of HMGB1, it is suggested that PDGF-B signal pathway plays a pivotal role in MS-induced HMGB1 acretylation and phosphorylation contributing VSMC release.

Taken together, our data suggested that MS induced HMGB1 secretion through an increased acetylation and phosphorylation of nuclear HMGB1 via PDGFR-β signaling. Thus, targeting the MS-PDGFR-β-HMGB1 axis in VSMCs might be a promising therapeutic strategy for vascular complications associated with hypertension.

## Supporting information

**S1 Raw images.**
(PDF)

**S2 Raw images.**
(PDF)

**S3 Raw images.**
(PDF)

**S4 Raw images.**
(PDF)

**S5 Raw images.**
(PDF)

## Author Contributions

**Conceptualization:** Seung Eun Baek.

**Data curation:** Ji On Kim, Seung Eun Baek.

**Formal analysis:** Ji On Kim.

**Investigation:** Ji On Kim.

**Methodology:** Ji On Kim.

**Project administration:** Seung Eun Baek.

**Resources:** Eun Yeong Jeon, Jong Min Choi, Eun Jeong Jang.

**Supervision:** Seung Eun Baek, Chi Dae Kim.

**Validation:** Ji On Kim.

**Visualization:** Ji On Kim.

Writing – **original draft:** Seung Eun Baek.

Writing – **review & editing:** Ji On Kim.

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
