## [Decision Letter · Decision Letter 0]

1 Dec 2021

PONE-D-21-21832PDGFR-β signaling mediates HMGB1 release in mechanically stressed vascular smooth muscle cellsPLOS ONE

Dear Dr. Kim,

Thank you for submitting your manuscript to PLOS ONE. After careful consideration, we feel that it has merit but does not fully meet PLOS ONE’s publication criteria as it currently stands. Therefore, we invite you to submit a revised version of the manuscript that addresses the points raised during the review process. The reviewers express concerns with the experimental design and result quality. the role of PDGFR-b mediate HMGB1 effects on smooth muscle cells should be included. Appropriate references needs to update and method section require more clarity.

Please submit your revised manuscript within 30 days. If you will need more time than this to complete your revisions, please reply to this message or contact the journal office at plosone@plos.org. Please include the following items when submitting your revised manuscript:A rebuttal letter that responds to each point raised by the academic editor and reviewer(s). You should upload this letter as a separate file labeled 'Response to Reviewers'.A marked-up copy of your manuscript that highlights changes made to the original version. You should upload this as a separate file labeled 'Revised Manuscript with Track Changes'.An unmarked version of your revised paper without tracked changes. You should upload this as a separate file labeled 'Manuscript'.

We look forward to receiving your revised manuscript.

Kind regards,

Suresh Kumar Verma, PhD

Academic Editor

PLOS ONE

Journal Requirements:

In your cover letter, please note whether your blot/gel image data are in Supporting Information or posted at a public data repository, provide the repository URL if relevant, and provide specific details as to which raw blot/gel images, if any, are not available. Email us at plosone@plos.org if you have any questions

Reviewers' comments:

Reviewer's Responses to Questions

**Comments to the Author**

1. Is the manuscript technically sound, and do the data support the conclusions?

Reviewer #1: No

Reviewer #2: Partly

Reviewer #3: Yes

2. Has the statistical analysis been performed appropriately and rigorously? 

Reviewer #1: I Don't Know

Reviewer #2: Yes

Reviewer #3: Yes

3. Have the authors made all data underlying the findings in their manuscript fully available?

Reviewer #1: Yes

Reviewer #2: Yes

Reviewer #3: Yes

4. Is the manuscript presented in an intelligible fashion and written in standard English?

Reviewer #1: No

Reviewer #2: No

Reviewer #3: Yes

5. Review Comments to the Author

Reviewer #1: In the present study, the authors studied signaling mediating release of HMGB1, a damage-associated molecular which mediates vascular complications in stress vasculature, in mechanically stressed vascular smooth muscle cells (VSMCs) isolated from mouse aorta SD rats. They found that HMGB1 released from stressed VSMCs was increased which accompanied with increased cytosolic translocation of nuclear HMGB1, and acetylated and phosphorylated form of HMGB1. Furthermore, they found that PDGFR inhibitor AG1295 inhibited HMGB1 release in stressed VSMCs. Moreover the increased HMGB1 release from stressed VSMCs was diminished when PDGFR-β was silenced by siRNA. PDGF-DD increased HMGB1 release. Finally the authors demonstrated that PDGFR-β-mediated secretion of HMGB1 in VSMCs might be a promising therapeutic strategy for hypertension-related vascular complications. The major concern of this manuscript is that the authors did not provide any evidence showing that the role of PDGFR-β/p-PDGFR-β-mediated HMGB1 in biology and function (i.e. contractile property) of VSMCs which makes the conclusion very weak. Other concerns including:

1. References should be added regarding the prerequisite of the cytosolic translation of nuclear HMGB1 for the secretion of HMGB1.

2. How p-PDGFR-β regulates HMGB1 cytosolic translocation and secretion?

3. What is the role of HMGB1 in regulation of stretch of VSMCs?

Reviewer #2: Dear Author,

I have received manuscript entitled “PDGFR-β signaling mediates HMGB1 release in mechanically stressed vascular smooth muscle cells; this is a good and significant article. In the present article author shown MS stressed VSMCs cells induced secreation hof HMGB1 via acetylation and phosphorylation of it. Also shown PDGFR-β pathways is critical for regulation of HMGB1 secretion during MS in VSMCs. Which further prove their hypothesis by knockdown and recombinant protein. This study is interesting, and manuscript can be considered for its publication. Nevertheless, I have few remarks concerning the manuscript. The manuscript can be considered for its publication after making the suggested changes.

Comments.

• NE-PERTM nuclear and cytoplasmic extraction, need to change with NE-PERTM

• Isolation of Endothelial cells of the excised ---------. Need to more elaborate. Is any enzymatic method also used for isolation?

• IP protocol need to elaborate. Also need to show pulldown with IgG as control.

• HMGB1 ELISA kit (Elabscience, Houston, TX, USA) mentions cat no. also.

• Statical analysis need to clearly mention about which group used ANOVA and in which group t test.

• How about HMGB1 mRNA expression does it also change in MS .

• Figure1 for mechanical stretch is not stimulation it should be cells stressed with Flexercell Tension Plus FX-4000T system.

• Figure 2, Run cytoplasmic and nuclear fraction into same gel and show pattern of housekeeping genes lamin B and actin both fractions which confirm your any cross contamination of fractions.

• Mention in each blot (Kda), size of proteins observed in immunoblot.

• Figure 3.In the IP experiment of acetylated- and phosphorylated HMGB1- need to show IgG as a negative control also.

• In the results section clearly mention after pulldown with HMGB1 immunoprobe with phospho or acetyl antibodies.

• Figure 4 HMGB1 secretion via PDGFR signaling in MS-stimulated VSMC, Mention concentration used for all inhibitors either Method section or in results.

• Figure 6, Knockdown efficiency does not clear, repeat the western and put clear significant change . Also add mRNA level of these genes in knockdown cells.

Reviewer #3: Kim et al studied the mechanism of vascular smooth muscle cell (VSMC) dysfunction under mechanical stress. The authors isolated VSMCs from rat thoracic aorta and cultured. When the VSMCs were subjected to 3% mechanical stress, an increase in the time-dependent cellular release of high motility group box 1 (HMGB1), a major damage associated molecular pattern, was observed. The mechanical stress also augmented the cytosolic translocation of nuclear HMGB1. Mechanical stress induced releases of HMGB1 was mediated by platelet-derived growth factor receptor (PDGFR), especially by PDGFR-beta signaling. This study reported a novel mechanism of VMSC dysfunction.

The manuscript is well written supporting a research experiment that was planned properly.

The authors discussed that this finding will be helpful to manage conditions such as hypertension, but smaller resistance vessels more role in the development of hypertension. Is it possible to duplicate the finding using VSMCs from smaller resistance vessels?

In the context of this experiment, interesting questions to be addressed by this group or any other group are:

1. whether AG1295, an inhibitor of PDGFR, can manage hypertension which is associated with mechanical stress of VSMCs?

2. What is the role of platelets under mechanical stress for the development of hypertension?

6. PLOS authors have the option to publish the peer review history of their article (what does this mean?). If published, this will include your full peer review and any attached files.

Reviewer #1: No

Reviewer #2: No

Reviewer #3: No

---

## [Author Response · Author response to Decision Letter 0]

15 Dec 2021

Reviewer #1: In the present study, the authors studied signaling mediating release of HMGB1, a damage-associated molecular which mediates vascular complications in stress vasculature, in mechanically stressed vascular smooth muscle cells (VSMCs) isolated from mouse aorta SD rats. They found that HMGB1 released from stressed VSMCs was increased which accompanied with increased cytosolic translocation of nuclear HMGB1, and acetylated and phosphorylated form of HMGB1. Furthermore, they found that PDGFR inhibitor AG1295 inhibited HMGB1 release in stressed VSMCs. Moreover the increased HMGB1 release from stressed VSMCs was diminished when PDGFR-β was silenced by siRNA. PDGF-DD increased HMGB1 release. Finally the authors demonstrated that PDGFR-β-mediated secretion of HMGB1 in VSMCs might be a promising therapeutic strategy for hypertension-related vascular complications. The major concern of this manuscript is that the authors did not provide any evidence showing that the role of PDGFR-β/p-PDGFR-β-mediated HMGB1 in biology and function (i.e. contractile property) of VSMCs which makes the conclusion very weak. Other concerns including:

As we absolutely agree with your comments, the following contents were included in the revised manuscript on page 14: “HMGB1 is known as a multifunctional protein which induces vascular remodeling in hypertension via phenotypic transformation of VSMCs from contractile to synthetic type (REF #31, 32, 33).”

1. References should be added regarding the prerequisite of the cytosolic translation of nuclear HMGB1 for the secretion of HMGB1.

The translocation of HMGB1 during cell activation is mediated by acetylation and phosphorylation processes, and is prerequisite for the secretion of HMGB1 (REF #27). These contents were included in the result section of the revised manuscript on page 9-10.

2. How p-PDGFR-β regulates HMGB1 cytosolic translocation and secretion?

Interestingly, the MS-induced HMGB1 secretion was significantly attenuated in PDGFR-β-deficient cells, but not in cells transfected with siRNA for PDGFR-α. Likewise, PDGF-DD, but not PDGF-AA, directly increased HMGB1 secretion in VSMCs, indicating a pivotal role of PDGFR-β signaling on MS-induced HMGB1 secretion in VSMCs. Considering our results in which MS increased HMGB1 release via acetylation and phosphorylation of HMGB1, it is suggested that PDGF-β signal pathway plays a pivotal role in MS-induced HMGB1 acretylation and phosphorylation contributing VSMC release. These contents were included in the revised manuscript on page 15.

3. What is the role of HMGB1 in regulation of stretch of VSMCs?

HMGB1 is known as a phenotypic modulator of VSMCs from contractile to synthetic type, might attenuates contractile property of VSMCs (REF #31, 32, 33). These contents were included in the revised manuscript on page 14.

 

Reviewer #2: Dear Author,

I have received manuscript entitled “PDGFR-β signaling mediates HMGB1 release in mechanically stressed vascular smooth muscle cells; this is a good and significant article. In the present article author shown MS stressed VSMCs cells induced secreation hof HMGB1 via acetylation and phosphorylation of it. Also shown PDGFR-β pathways is critical for regulation of HMGB1 secretion during MS in VSMCs. Which further prove their hypothesis by knockdown and recombinant protein. This study is interesting, and manuscript can be considered for its publication. Nevertheless, I have few remarks concerning the manuscript. The manuscript can be considered for its publication after making the suggested changes.

Comments.

• NE-PERTM nuclear and cytoplasmic extraction, need to change with NE-PERTM

The revised manuscript has been revised as your indications on Page 5.

• Isolation of Endothelial cells of the excised ---------. Need to more elaborate. Is any enzymatic method also used for isolation?

In our primary culture process for VSMCs, endothelial cells were not isolated but just physically peeled off with stainless steel scraper, and any enzymatic method was not used. These contents were included in the revised manuscript on Page 6.

• IP protocol need to elaborate. Also need to show pulldown with IgG as control.

To verify whether the antibody only affects target protein, HMGB1, we evaluated positive control (IP input) instead of negative control (IgG), as shown in other previous report by Xu et al. (Front Physiol. 2019, 5;10:854. doi: 10.3389/fphys.2019.00854). The procedures for IP were thoroughly revised in the revised manuscript on Page 7.

• HMGB1 ELISA kit (Elabscience, Houston, TX, USA) mentions cat no. also.

Manuscript has been thoroughly revised as your indications.

• Statical analysis need to clearly mention about which group used ANOVA and in which group t test.

Manuscript has been thoroughly revised as your indications.

• How about HMGB1 mRNA expression does it also change in MS .

In our study, we investigated the regulatory signals involved HMGB1 secretion in MS-stimulated VSMCs, but not transcriptional regulation of HMGB1 synthesis as shown in other paper (Volchuk A et al. Nat Commun. 2020 Sep 11;11(1):4561. doi: 10.1038/s41467-020-18443-3). Thus, we focussed on the translocation and secretion of HMGB1 in VSMCs stimulated by MS.

• Figure1 for mechanical stretch is not stimulation it should be cells stressed with Flexercell Tension Plus FX-4000T system.

Manuscript has been changed as your indications (Page 27).

• Figure 2, Run cytoplasmic and nuclear fraction into same gel and show pattern of housekeeping genes lamin B and actin both fractions which confirm your any cross contamination of fractions.

Unfortunately, we followed the previous method showing sphingosine kinase 1-regulated translocation of HMGB1 by Tian T et al. (Cell Death Dis. 2020;11(12):1037. doi: 10.1038/s41419-020-03255-6u I). In the next study, we sincerely hope to run cytoplasmic and nuclear fraction into same gel.

• Mention in each blot (Kda), size of proteins observed in immunoblot.

All of the figures have been revised as your indications.

• Figure 3.In the IP experiment of acetylated- and phosphorylated HMGB1- need to show IgG as a negative control also.

To verify whether the antibody only affects target protein, HMGB1, we evaluated positive control (IP input) instead of negative control (IgG), as shown in other previous report by Xu et al. (Front Physiol. 2019, 5;10:854. doi: 10.3389/fphys.2019.00854). 

• In the results section clearly mention after pulldown with HMGB1 immunoprobe with phospho or acetyl antibodies.

Manuscript has been revised as your indications on Page 10.

• Figure 4 HMGB1 secretion via PDGFR signaling in MS-stimulated VSMC, Mention concentration used for all inhibitors either Method section or in results.

The concentrations used for all inhibitors for Figure 4 were included in the figure legends in the revised manuscript on Page 28.

• Figure 6, Knockdown efficiency does not clear, repeat the western and put clear significant change . Also add mRNA level of these genes in knockdown cells.

As we absolutely agree with your indications, Western blot data showing clear changes (Figure 6 in the revised manuscript) were included in the revised manuscript.

 

Reviewer #3: Kim et al studied the mechanism of vascular smooth muscle cell (VSMC) dysfunction under mechanical stress. The authors isolated VSMCs from rat thoracic aorta and cultured. When the VSMCs were subjected to 3% mechanical stress, an increase in the time-dependent cellular release of high motility group box 1 (HMGB1), a major damage associated molecular pattern, was observed. The mechanical stress also augmented the cytosolic translocation of nuclear HMGB1. Mechanical stress induced releases of HMGB1 was mediated by platelet-derived growth factor receptor (PDGFR), especially by PDGFR-beta signaling. This study reported a novel mechanism of VMSC dysfunction.

The manuscript is well written supporting a research experiment that was planned properly.

The authors discussed that this finding will be helpful to manage conditions such as hypertension, but smaller resistance vessels more role in the development of hypertension. Is it possible to duplicate the finding using VSMCs from smaller resistance vessels?

Although VSMCs in smaller resistance vessels play important role in blood pressure regulation, we can hardly culture VSMCs from smaller resistance vessels. Thus, we could not perform an experiment using VSMCs in smaller resistance vessels. However, we think that there is high probability of the same result in a smaller resistance vessel.

In the context of this experiment, interesting questions to be addressed by this group or any other group are:

1. whether AG1295, an inhibitor of PDGFR, can manage hypertension which is associated with mechanical stress of VSMCs?

In our present study, we investigated HMGB1 release in mechanically stressed VSMCs, but not contractile regulation of VSMCs. Considering the previous reports describing the role of HMGB1 on vascular complications, it is suggested that PDGFR inhibitor might be useful to prevent vascular complication caused by hypertension.

2. What is the role of platelets under mechanical stress for the development of hypertension?

Considering the previous reports describing the role of platelet activation in hypertension (Griffin G, Am Fam Physician, 2005;71(5):897-9; Gkaliagkousi E et al, Am J Hypertens, 2010;23(3):229-36; El Haouari M et al, Blood Cells Mol Dis, 2009;42(1):38-43), it is suggested that platelet might play an important role under mechanical stress for the development of hypertension. Considering the facts that PDGF was released from platelets (Linder BL et al, Proc Natl Acad Sci U S A, 1979;76(8):4107-11), future experiments are needed to determine the active role of PDGF under mechanical stress for the development of hypertension.

---

## [Decision Letter · Decision Letter 1]

28 Feb 2022

PDGFR-β signaling mediates HMGB1 release in mechanically stressed vascular smooth muscle cells

PONE-D-21-21832R1

Dear Dr. Kim,

We’re pleased to inform you that your manuscript has been judged scientifically suitable for publication and will be formally accepted for publication once it meets all outstanding technical requirements.

Kind regards,

Suresh Kumar Verma, PhD

Academic Editor

PLOS ONE

Additional Editor Comments (optional):

Reviewers' comments:

Reviewer's Responses to Questions

**Comments to the Author**

1. If the authors have adequately addressed your comments raised in a previous round of review and you feel that this manuscript is now acceptable for publication, you may indicate that here to bypass the “Comments to the Author” section, enter your conflict of interest statement in the “Confidential to Editor” section, and submit your "Accept" recommendation.

Reviewer #2: All comments have been addressed

Reviewer #4: All comments have been addressed

2. Is the manuscript technically sound, and do the data support the conclusions?

Reviewer #2: Partly

Reviewer #4: Yes

3. Has the statistical analysis been performed appropriately and rigorously? 

Reviewer #2: I Don't Know

Reviewer #4: Yes

4. Have the authors made all data underlying the findings in their manuscript fully available?

Reviewer #2: Yes

Reviewer #4: Yes

5. Is the manuscript presented in an intelligible fashion and written in standard English?

Reviewer #2: (No Response)

Reviewer #4: Yes

6. Review Comments to the Author

Reviewer #2: Title PDGFR-β signaling mediates HMGB1 release in mechanically stressed vascular

smooth muscle cells, Baek etal addressed my comments. Don't have any further

Reviewer #4: In the revised manuscript, the authors have adequately addressed the reviewer's comments. It can be published from a scientific point of view.

7. PLOS authors have the option to publish the peer review history of their article (what does this mean?). If published, this will include your full peer review and any attached files.

Reviewer #2: No

Reviewer #4: **Yes: **Prabhat Ranjan

---

## [Editor Report · Acceptance letter]

8 Mar 2022

PONE-D-21-21832R1 

PDGFR-β signaling mediates HMGB1 release in mechanically stressed vascular smooth muscle cells 

Dear Dr. Kim:

I'm pleased to inform you that your manuscript has been deemed suitable for publication in PLOS ONE. Congratulations! Your manuscript is now with our production department. 

Kind regards, 

on behalf of

Dr. Suresh Kumar Verma 

Academic Editor

PLOS ONE